# Secretory quality control constrains functional selection-associated protein structure innovation

Bin Cheng[1,5], Jian-Min Lv[2,5], Yu-Lin Liang[1], Li Zhu [1], Xiao-Ping Huang[1], Hai-Yun Li[2], Lawrence A. Potempa[3], Shang-Rong Ji [1✉] & Yi Wu [2,4✉]

Biophysical models suggest a dominant role of structural over functional constraints in shaping protein evolution. Selection on structural constraints is linked closely to expression levels of proteins, which together with structure-associated activities determine in vivo functions of proteins. Here we show that despite the up to two orders of magnitude differences in levels of C-reactive protein (CRP) in distinct species, the in vivo functions of CRP are paradoxically conserved. Such a pronounced level-function mismatch cannot be explained by activities associated with the conserved native structure, but is coupled to hidden activities associated with the unfolded, activated conformation. This is not the result of selection on structural constraints like foldability and stability, but is achieved by folding determinants-mediated functional selection that keeps a confined carrier structure to pass the stringent eukaryotic quality control on secretion. Further analysis suggests a folding threshold model which may partly explain the mismatch between the vast sequence space and the limited structure space of proteins.

[1] MOE Key Laboratory of Cell Activities and Stress Adaptations, School of Life Sciences, Lanzhou University, Lanzhou, P.R. China. [2] MOE Key Laboratory of Environment and Genes Related to Diseases, School of Basic Medical Sciences, Xi'an Jiaotong University, Xi'an, P.R. China. [3] Roosevelt University College of Pharmacy, Schaumburg, IL 60173, USA. [4] Key Laboratory of Precision Medicine to Pediatric Diseases of Shaanxi Province, Xi'an Children's Hospital, Xi'an Jiaotong University, Xi'an, P.R. China. [5] These authors contributed equally: Bin Cheng, Jian-Min Lv. ✉email: jsr@lzu.edu.cn; wuy@lzu.edu.cn

The mechanism of protein evolution is a fundamental question[1,2]. Functional importance was initially thought to be the primary constraint on protein evolution. However, subsequent theoretical and empirical studies unexpectedly revealed that expression level[3], but not functional importance[4] is the major determinant. Expression level per se, though contributing explicitly to protein function, is also linked closely to selection on structural constraints to avoid misfolding[5,6] or misinteraction[7,8]. Moreover, the site-specific evolution within proteins provides further support to a dominant role of structural over functional constraints[2,9]. Indeed, besides enzymes[10–12], most conserved sites of proteins are directly related to stability but not to functionality[13–16]. Case studies on reconstructed or experimental trajectories of protein evolution also highlighted critical contributions from structural constraints like foldability[17,18].

However, the contributions of functional constraints to protein evolution might be underestimated in previous studies due to several practical difficulties. First, functional constraints should be manifested by the conserved, improved or adapted in vivo functions of a protein during evolution. This cannot be weighted simply by effects of gene knockout (KO) on cell survival or growth of a single species, or be measured by changes of in vitro protein activities across species. Second, the in vivo functions of a protein may rely on additional (albeit unknown) conformations that are less considered in biophysical modeling. Third, expression levels critically affect in vivo functions of proteins, yet their species-specific differences are usually ignored. Regarding these difficulties, C-reactive protein (CRP), a clinical marker of inflammation, might represent a unique model to evaluate the relative contributions of structural versus functional constraints during evolution.

CRP belongs to the highly conserved short-chain pentraxin family with calcium-dependent ligand-binding activities[19–21]. CRP is presumed to act as a hepatocyte-secreted pattern recognition receptor that binds phosphorylcholine (PC) exposed to invading pathogens or damaged cell membranes[22,23]. In most species, CRP is composed of five identical, non-covalently assembled subunits whose tertiary structure keeps nearly unchanged during evolution[19,21,24]. In contrast, the expression pattern and circulating level of CRP manifest strong variation across species[19,21]. For example, human CRP is a major acute-phase protein whose circulating level can increase up to 1000-fold, from the baseline of <1 µg/ml to >500 µg/ml upon infection or tissue injury[19–21]. However, mouse and rat CRP are at best minor acute phase proteins with their baseline levels being 7.5 and 300 µg/ml, respectively[19,21,25].

Such a level-structure mismatch immediately raises a critical concern on the functional importance or conservation of CRP, which would dictate how mice and rats, the most widely used animal models, should be manipulated to define the in vivo actions of human CRP[19,26]. Moreover, if indeed CRP possesses important functions as implicated by its conserved sequence and structure, then what underlies the functional adaptation to the drastically changed circulating levels of CRP during evolution? Here we demonstrate that the in vivo functions of CRP are essential and conserved across species, with the level-structure mismatch reconciled by functional selection on hidden activities expressed only in unfolded CRP encompassing conformational states denoted as mCRP, mCRPm, and pCRP* in literatures[27]. We further propose that such a functional selection must be confined to a conserved native structure due to the stringent quality control on secretion in eukaryotic cells.

## Results

### In vivo functions of CRP in acute inflammation are conserved.
The drastic differences between circulating levels of mouse, rat, and human CRP suggest that their in vivo functions are either distinct or redundant[19,26]. To assess this suggestion, we compared the functional phenotypes of mouse and rat CRP KO in two acute inflammatory diseases, i.e., acetaminophen-induced liver failure and lipopolysaccharide (LPS)-induced sepsis. CRP KO markedly aggravated (Fig. 1a, b), whereas human CRP injection effectively alleviated the severity of both diseases (Fig. 1c). These results thus reveal a conserved and nonredundant function of CRP in protecting against tissue injury caused by acute inflammation. Moreover, previous studies including ours have also demonstrated conserved and nonredundant functions of CRP in renal ischemia-reperfusion injury (IRI)[28], diabetic nephropathy (DKD)[29,30], and collagen-induced arthritis (CIA)[31] by using CRP KO or human CRP transgenic mice and CRP KO rats.

**Hidden activities of unfolded CRP match in vivo functions.** Given their drastic differences in circulating levels, the conserved in vivo function of mouse, rat, and human CRP would otherwise

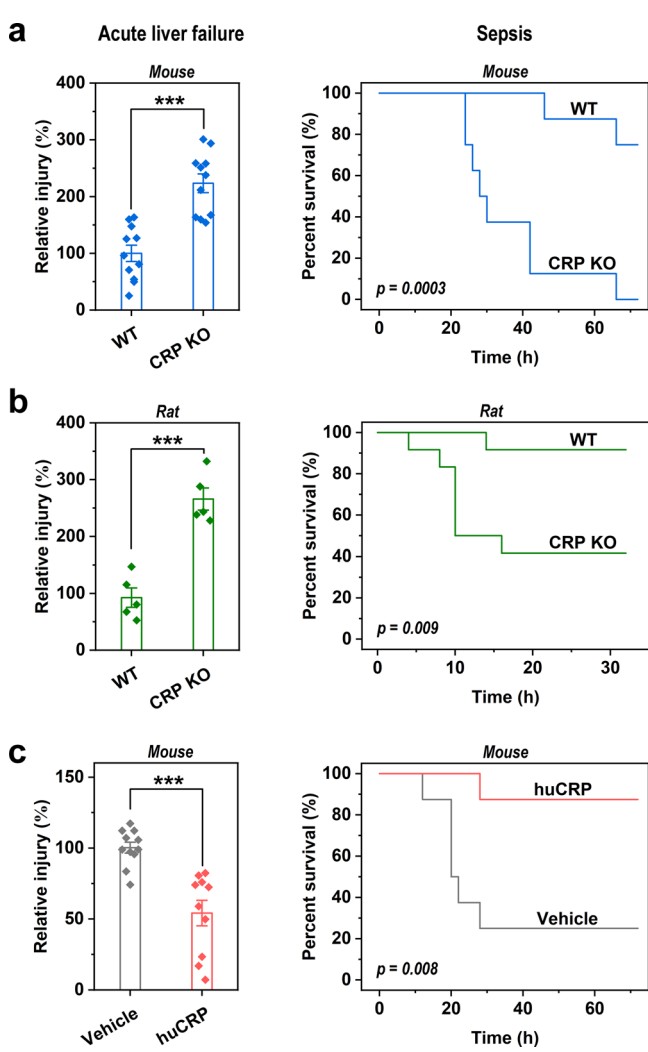

**Fig. 1 In vivo functions of CRP are evolutionarily conserved.** Acute liver failure ($n = 11$ mice/group; $n = 5$ rats/group) and sepsis ($n = 8$ mice/group; $n = 12$ rats/group) were induced in wild-type and CRP KO mice (**a**) or rats (**b**) by i.p. injection of acetaminophen or LPS. **c** Human CRP (huCRP) was administrated into wild-type mice with acute liver failure ($n = 11$ for vehicle; $n = 10$ for huCRP treatment) or sepsis ($n = 10$ mice/group). CRP KO aggravated, whereas huCRP administration alleviated both diseases. These results reveal consistent in vivo functional phenotypes of mouse, rat, and human CRP. Data are presented as mean ± SEM; ***$p < 0.001$, two-tailed Student's $t$ test, two-sided.

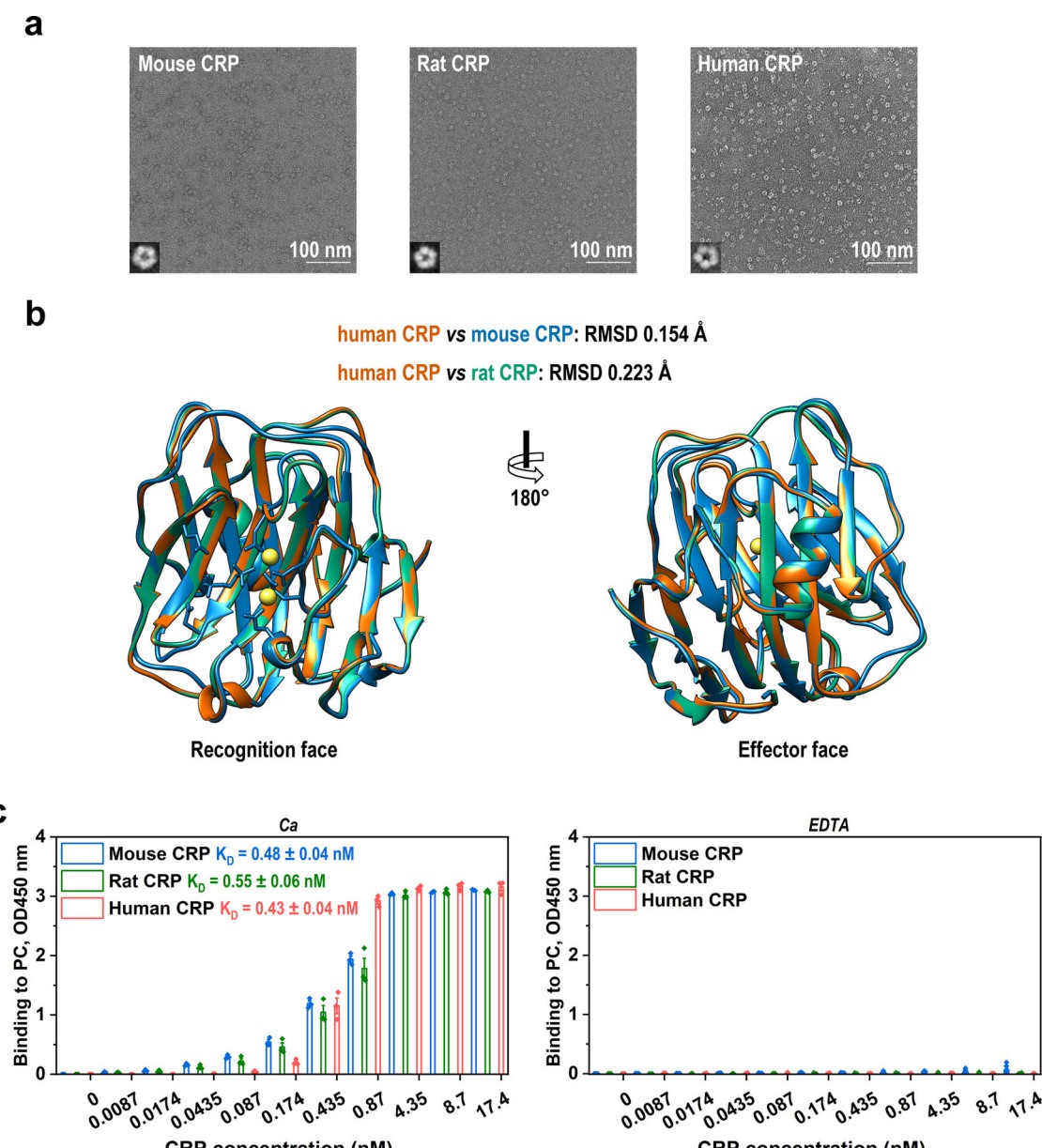

**Fig. 2 The native structure and associated activities of CRP are evolutionarily conserved. a** The pentameric assembly of purified mouse, rat, and human CRP was examined with electron microscopy and single-particle analysis. The diameters of pentameric class averages of the mouse (181 particles), rat (209 particles), and human CRP (208 particles) are 10.98, 10.97, and 10.24 nm, respectively. **b** The subunit structures of mouse and rat CRP were generated with SWISS-MODEL using subunit A from the crystal structure of human CRP (PDB 1B09)[56] as the template. **c** The binding of mouse, rat, and human CRP to immobilized PC-KLH were examined with ELISA in the presence (left; n = 3 independent experiments) or absence of calcium (right; n = 6 independent experiments for mouse CRP; n = 3 independent experiments for rat and human CRP). Neither their native structures nor calcium-dependent PC-binding activities exhibited significant difference. Data are presented as mean ± SEM.

suggest distinct activities to explain the level-function mismatch. As activities are determined by structure, we first characterized the structural states of the mouse, rat, and human CRP. Little alterations in their pentameric assembly (Fig. 2a) and subunit conformation (Fig. 2b), however, were noted by electron microscopy visualization, single particle analysis, and homology modeling. In vitro PC binding is an established activity of CRP closely associated with its in vivo function[22,23]. However, mouse, rat, and human CRP also showed a comparable capacity to bind PC in a calcium-dependent manner (Fig. 2c). These results together demonstrate that the native structures and associated activities of mouse, rat, and human CRP are comparable, which therefore cannot explain the pronounced level-function mismatch.

We and others have shown that binding of CRP to PC exposed on damaged cell membranes leads to pentamer dissociation and subunit unfolding, eventually forming monomeric CRP (mCRP) that exhibits greatly enhanced activities[32,33]. Interestingly, mouse CRP was most, whereas rat CRP was least sensitive to heat- (Fig. 3a) and urea-induced unfolding (Fig. 3b). After removal of urea, however, unfolded rat CRP underwent efficient refolding, while unfolded mouse and human CRP did not (Fig. 3c). Therefore, mouse CRP is most, whereas rat CRP is least prone to forming mCRP. Moreover, the major functional motif of mCRP, i.e., cholesterol-binding sequence (CBS; a.a. 35–47)[32,34–37], was also most active for mouse but was least active for rat (Fig. 3d). These results together argue that level variations of CRP are

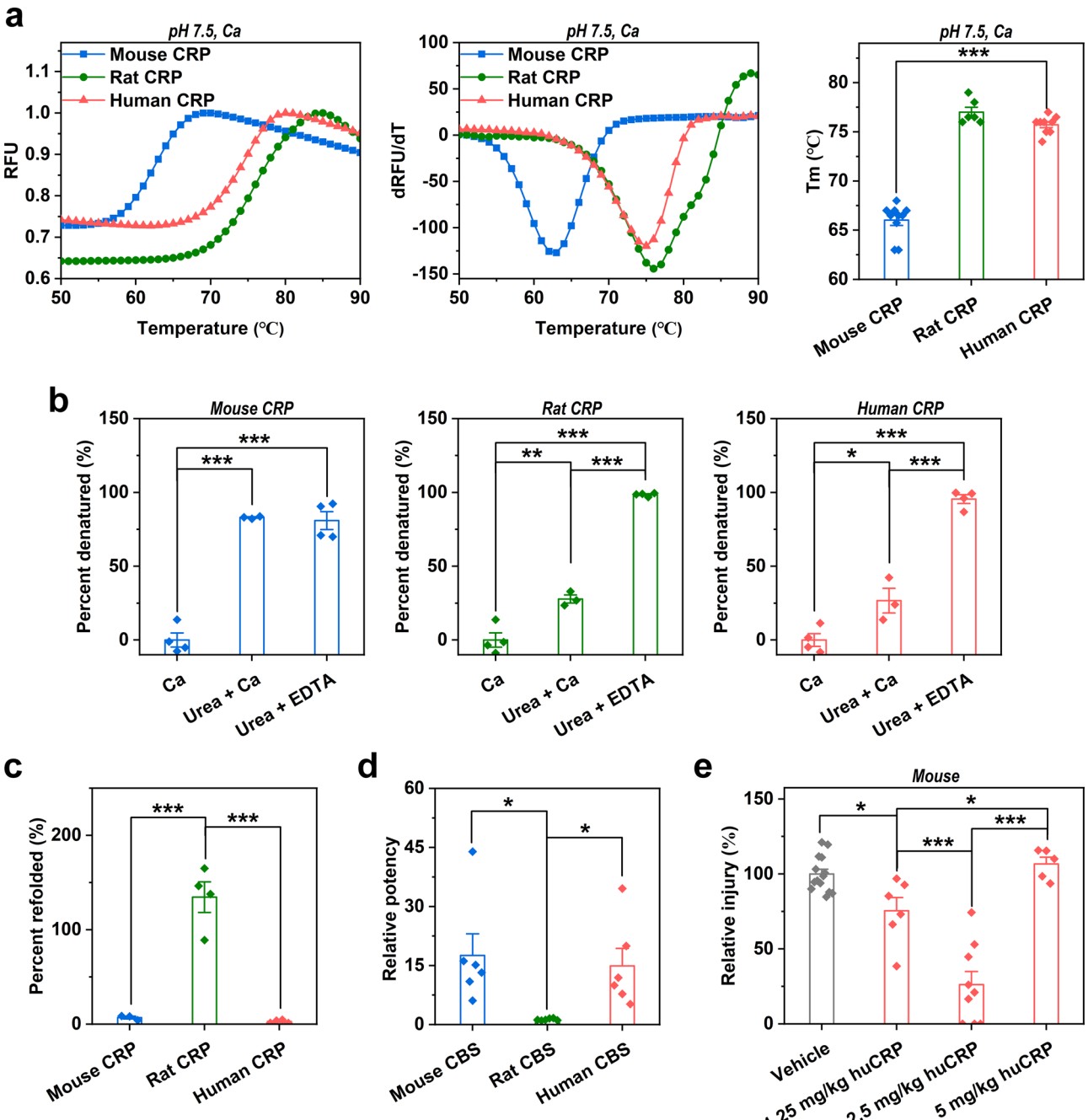

**Fig. 3 The unfolded structure and associated hidden activities of CRP are evolutionarily distinct.** The stability of mouse, rat, and human CRP was examined with heat- (**a**) ($n = 10$ independent experiments for mouse CRP; $n = 6$ independent experiments for rat CRP; $n = 9$ independent experiments for human CRP) and urea-induced unfolding (**b**) ($n = 4$ independent experiments). The melting temperature (Tm) of mouse CRP was much lower than that of rat and human CRP. Accordingly, calcium exhibited little effect on urea-induced unfolding of mouse CRP, but markedly inhibited that of rat and human CRP. **c** The refolding of urea-unfolded CRP was examined following dilution ($n = 3$ independent experiments for mouse CRP; $n = 4$ independent experiments for rat and human CRP). Rat CRP efficiently refolded, whereas mouse and human CRP did not. **d** The hidden activities of the major functional motif of unfolded CRP, i.e., CBS, were examined with competitive ELISA against 6 ligands[36]. Relative potency for each ligand was defined as: binding of mCRP in the absence of CBS divided by that in the presence of CBS. Each data point represents a potency index of a different ligand ($n = 3$ independent experiments). Mouse CBS was most active, while rat CBS was least active. These results indicate that mouse CRP is most prone to unfold and possesses the strongest hidden activities. By contrast, rat CRP is most resistant to unfolding and possesses the weakest hidden activities. **e** Wild-type mice with acetaminophen-induced liver failure were treated with human CRP at the indicated dosages ($n = 14$ mice for vehicle; $n = 6$ mice for 1.25 mg/kg dosage; $n = 9$ mice for 2.5 mg/kg dosage; $n = 5$ mice for 5 mg/kg dosage). Human CRP reduced liver injury at dosages of 1.25 and 2.5 mg/kg, but showed little effect at the dosage of 5 mg/kg. The loss of protection of human CRP at the highest dosage might be due to strong proinflammatory responses evoked by excessive mCRP. Data are presented as mean ± SEM; *$p < 0.05$; **$p < 0.01$; ***$p < 0.001$, one-way ANOVA with Tukey post hoc.

matched by the ease of mCRP generation and the associated activities to manifest consistent in vivo functions.

As the levels of CRP and the generation/activities of mCRP are negatively coupled across species, achieving a proper range of levels should be essential to correctly define the functions of human CRP in animal models. Indeed, at dosages of 1.25 and 2.5 mg/kg, human CRP protected wild-type (WT) mice against acetaminophen-induced liver injury in a dose-dependent manner (Fig. 3e), consistent with the functional phenotype of mouse CRP KO. At the dosage of 5 mg/kg, however, human CRP was no longer protective (Fig. 3e). This may be ascribed to the excessive generation or activities of mCRP, which would evoke proinflammatory responses from neutrophiles and macrophages[27], eventually negating its protective actions, such as inhibiting complement overactivation[37,38]. Therefore, avoiding the detrimental effects caused by excessive generation or activities of mCRP appears to be the major constraint on CRP evolution.

**Structural constraints play a minor role in shaping CRP evolution**. The way exploited by CRP to keep its conserved in vivo functions suggests a minor role of structural constraints in shaping CRP evolution. To assess this suggestion, we selected 11 nodes of CRP along the evolutionary trajectory (Fig. 4a) and characterized their foldability, i.e., the efficiency to form the native structure[17], and stability in E. coli cells to take account of both biophysical properties of CRP itself and inevitable influences posed by cellular factors, e.g., chaperones[17,18]. The foldability of CRP was evaluated by soluble expression and PC binding activity in cell lysates, which reflect foldable and (nearly) folded CRP within cells, respectively, and by PC binding activity in culture media, which reflects fully folded and secreted CRP[17,39]. The stability of CRP was evaluated by residual PC binding activity in cell lysates and culture media after heat- or urea-induced unfolding[17]. None of these parameters, however, showed an apparent evolutionary correlation among the 11 nodes of CRP (Fig. 4b–g), demonstrating a minor role of structural constraints.

**Folding determinants dictate the native structure of CRP**. If structural constraints do not dominate CRP evolution, then it is somewhat paradoxical that selection on hidden activities associated with unfolded CRP is confined to a nearly identical native structure. For example, the sequence identity between zebrafish and human CRP is about 32%, but their X-ray crystal structures differ by only 0.73 Å of RMSD per subunit[24]. To understand such an evolutionary pattern, we calculated conservation scores at each position of CRP and mapped these scores to the X-ray crystal structure of human CRP[40]. It then became apparent that functional sites of native CRP, i.e., PC-binding site on the recognition face and the C1q- and FcγR-binding sites on the effector face, are significantly more variable than others (Fig. 5a). Importantly, mutating these sites does not affect the native structure formation (folding) of CRP in mammalian cells[39,41–44]. Therefore, residues critical to activities of native CRP are evolved to segregate from those critical to its folding.

We have previously shown that the cellular folding of CRP depends on the formation of the hydrophobic core consisting of β strands A to M (Supplementary Fig. 1)[39]. In this two-stepped process, strands C to I fold spontaneously first, leading to the bonding between Cys36 and Cys97, which upon calcium binding drives the subsequent nonspontaneous integration of strands J and K to finalize the assembly of the resting strands (Supplementary Fig. 1). Interestingly, mutating residues within strands C to I showed at best moderate impact on its spontaneous folding, while mutating residues critical to disulfide bonding or calcium binding abrogated the subsequent nonspontaneous folding

regardless of the presence of cellular factors (Fig. 5b). As those folding-decisive residues are specific to the folded structure, their highly conserved trait (Fig. 5a, c) suggests that they likely represent folding determinants evolved to keep the native structure of CRP minimally changed during evolution.

**Folding determinants constitute functional constraints of CRP**. Besides dictating the folded structure, the folding determinants of CRP are also tightly coupled to functional selection. First, mutating folding determinants precludes the secretion of CRP by mammalian cells (Fig. 5d)[39,42], likely due to the stringent eukaryotic quality control[45]. This is functionally equivalent to CRP KO which has not been observed in mammals likely due to impairment of fitness[19,21]. By contrast, the formation of inclusion bodies and efficient secretion of those mutants could still be observed in the prokaryotic E. coli cells whose quality control is less stringent. Second, Cys36 is located within CBS, whose activities are therefore hidden in native CRP by both spatial packing and disulfide locking (Fig. 5e). Third, the flexible sequence pattern of CBS (**L/V**-X(1-5)-**Y**-X(1-6)-**R/K**)[34] nevertheless permits mutations to fine-tune activities of unfolded CRP to match level variations (Fig. 5f). These results thus suggest that folding determinants represent a major part of functional constraints that dominate the evolution of CRP.

Because the strongest consequence of selection against folding determinants of CRP, i.e., functional KO, appears to be imposed by cellular quality control, it is possible that such a mode might also apply to other secretory proteins. To gain support to this hypothesis, we examined an extreme case of folding, i.e., intrinsically disordered protein (IDP). IDPs are unable to fold spontaneously into well-defined 3D structures[46], and therefore are less likely to pass the stringent quality control required for secretion. Indeed, analysis of DisProt database[47] revealed that the ratio of secretory IDPs is <0.15 in all analyzed eukaryotic organisms (Fig. 5g). Moreover, disordered sequences are drastically enriched in cytoplasmic portions of membrane proteins regardless of the organelles they localize (Fig. 5h). These would further suggest that luminal proteins might be similarly controlled as secretory proteins. We thus propose a folding threshold model wherein functional constraints-dominated protein evolution occurs in a confined carrier structure due to the stringent eukaryotic quality control (Fig. 6).

## Discussion

The present study has demonstrated an essential in vivo function of CRP in acute inflammation that is conserved across species regardless of the evolutionary variation in levels. Our findings, therefore, argue that mice and rats should not be regarded as natural models with their endogenous CRP defective in expression or function as assumed previously[19,26]. Instead, these animals can be manipulated appropriately to define the function of human CRP. Appropriate manipulations should include KO of endogenous CRP and rescue with WT or mutant human CRP at proper levels. Importantly, it is the hidden activities associated with unfolded CRP, rather than the native activities associated with folded CRP, that match the varied levels to keep the conserved in vivo function. This, in combination with the identification of autoantibodies specific to unfolded CRP in a human disease[37,48], its detection with conformation specific-antibodies in vivo and in patient tissues[49–53], and markedly enhanced activities[27], provides unambiguous support for the functional importance of unfolded CRP or mCRP in vivo.

The functional importance of unfolded CRP also raises an interesting question that why CRP has to keep its native structure during evolution. It has been proposed that native CRP represents

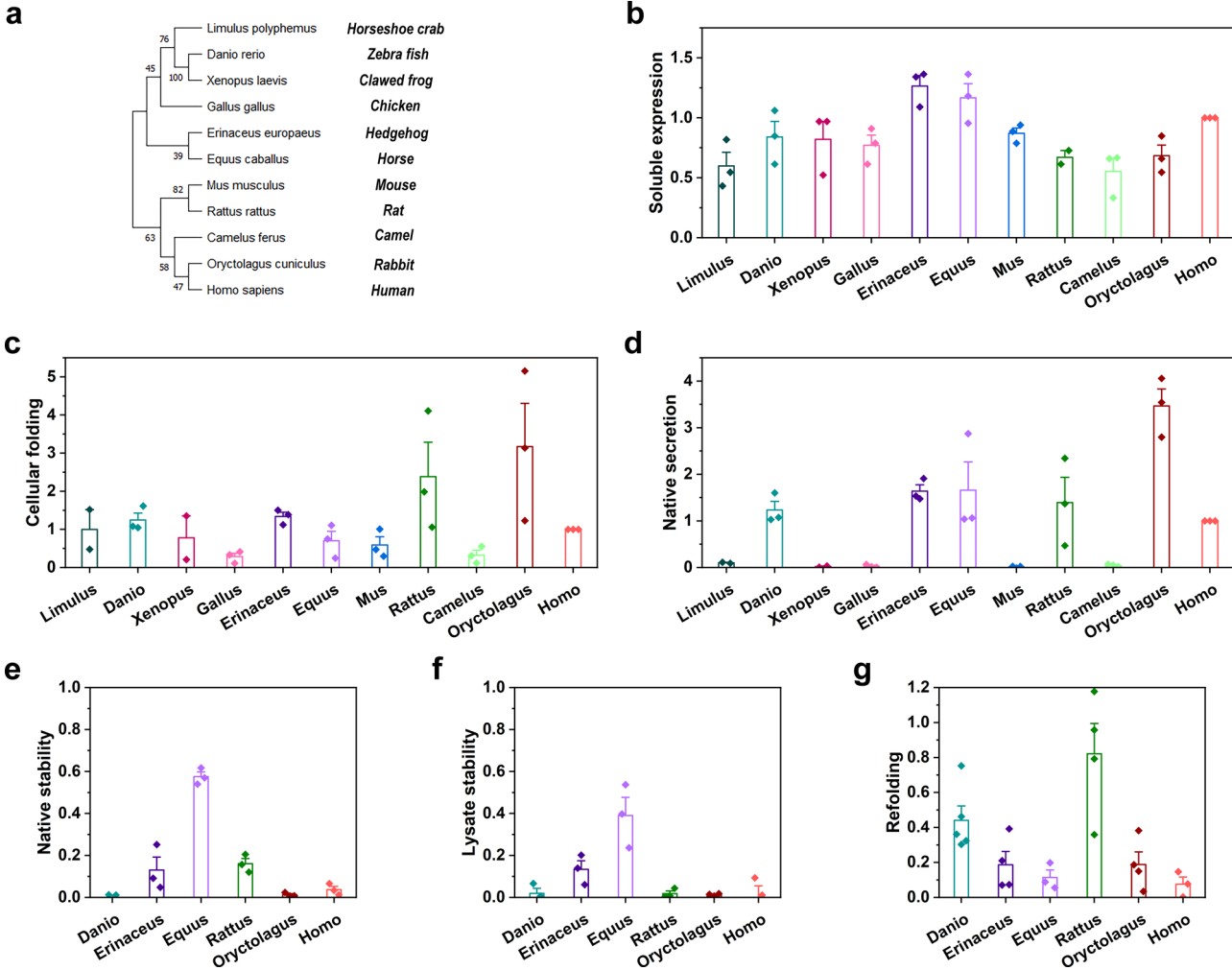

**Fig. 4 Structural constraints show little effect in shaping CRP evolution. a** Phylogenetic tree of CRP amino acid sequences from 11 species was constructed by the Neighbor-Joining method using MEGA-X[65]. The percentage of replicate trees in which the associated taxa clustered together in the bootstrap test (500 replicates) are shown next to the branches. The tree is drawn to scale, with branch lengths in the same units as those of the evolutionary distances used to infer the phylogenetic tree. CRP of 11 species was expressed in *E. coli* cells, and their expression levels in supernatants of cell lysates ($n = 3$ independent experiments) (**b**), PC-binding activities in supernatants of cell lysates ($n = 2$ independent experiments for Limulus and Xenopus; $n = 3$ independent experiments for other species) (**c**), and culture media ($n = 2$ independent experiments for Limulus and Xenopus; $n = 3$ independent experiments for other species) (**d**) were determined with immunoblotting or ELISA. The obtained values were then normalized to that of human CRP. CRP of 6 species with extracellular PC-binding activities was further examined with their thermal stability in culture media ($n = 3$ independent experiments) (**e**) and cell lysates ($n = 3$ independent experiments) (**f**) upon heating to 70 °C, and their capacities to refold in culture media after urea treatment ($n = 3$ independent experiments for Equus and Homo; $n = 4$ independent experiments for other species) (**g**). These measured parameters showed little evolutionary correlation among the 11 nodes of CRP. Data are presented as mean ± SEM.

a carrier releasing unfolded CRP only when reaching the site of injury[54]. Indeed, activities associated with the native structure either are related to the generation of mCRP, i.e., binding to PC exposed on damaged cell membranes[32], or can be performed also by mCRP, i.e., complement activation[38]. Moreover, large evolutionary changes are not unusual for CRP as evidenced by the presence of another short-chain pentraxin (i.e., serum amyloid P component), by the presence of structurally reorganized, long-chain pentraxins, by the presence of multiple CRP isoforms in crustaceans and lower vertebrates, and by the presence of a pseudogene of CRP in mammals[19–21,55]. Even so, CRP still manifests a delicate evolutionary pattern wherein adjustments to the generation and/or activities of the unfolded structure are selected only if the native structure is minimally altered.

That might be due to the stringent eukaryotic quality control that preferentially licenses well-folded structures for secretion, the failure of which is equivalent to functional KO for a secretory protein. In the case of human CRP, no stable, well-folded conformations other than the (near) native structure have been detected during its stepwise folding in live cells and unfolding in vitro[39]. This would suggest that only one folded state is encoded by the sequence of human CRP. The formation of this state critically depends on a handful of residues, i.e., the folding determinants, whose mutation abrogates the secretion of human CRP in mammalian cells. Mutations outside the folding determinants, however, are highly tolerable, as known native structures of CRP orthologs and paralogs are very similar despite their extensive sequence variations[24,56]. Therefore, mutations within CRP either disrupt or retain the original structure with a low possibility to create a stable alternative structure capable of passing the eukaryotic quality control on secretion.

As such, the highly conserved folding determinants serve as the primary functional constraint by keeping the native structure of CRP to secure its secretion and function. This sets the context for the

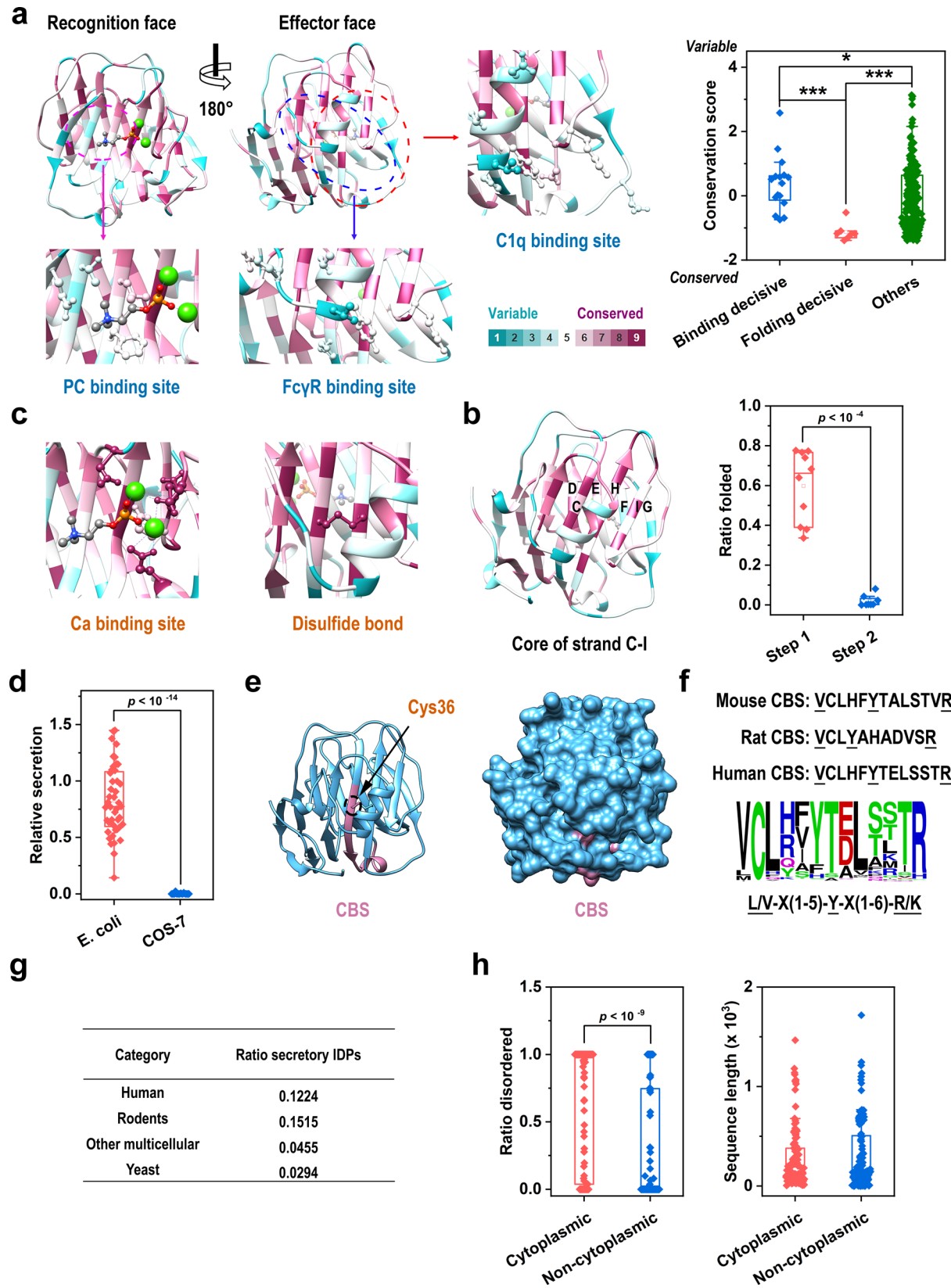

secondary adjustment to the generation or activities of mCRP to match levels of CRP. Though lacking unambiguous evidence, we reason that level changes of CRP are primarily due to noncoding mutations within regulatory elements and might occur earlier than the corresponding adjustments to the hidden activities. First, the promoter that controls *CRP* expression[19,21] is much less conserved than the coding sequence (Supplementary Figure 2). Second, the drastic difference in circulating levels and acute phase responses of mouse, rat, and human CRP cannot be explained by coding mutations. Third, noncoding mutations are usually more tolerable and milder, permitting not only their progressive accumulation during evolution to generate stronger impact but also gradual adaptation.

**Fig. 5 Folding determinants shape the evolutionary pattern of CRP.** Conservation scores of CRP were calculated with Consurf[40] using 121 available amino acid sequences of different species. **a** Conservation scores were mapped onto the crystal structure of the human CRP subunit (PDB 1B09)[56]. Side chains of residues critical to PC-, C1q-, and FcγR-binding are shown (binding-decisive) (left), and their conservation scores are compared with scores of other CRP residues (right). **b** Strands of the hydrophobic core that folds spontaneously[39] are indicated (left). The effects of mutations on CRP folding were evaluated (right). **c** Side chains of residues critical to calcium binding and disulfide bonding (folding-decisive) were shown with mapped conservation scores. Binding-decisive residues were significantly more variable than folding-decisive residues and those not directly involved in binding and folding (others). Mutating residues within the hydrophobic core did not abrogate its spontaneous formation (Step 1; 10 mutants examined, Supplementary Table 1), whereas mutating residues involved in disulfide bonding or calcium binding abrogated the subsequent nonspontaneous folding (Step 2; 8 mutants examined, Supplementary Table 1). Each data point represents the relative folding of a different mutant ($n = 3$ independent experiments). **d** The effects of mutations at folding determinants on CRP secretion by prokaryotic E. coli (44 mutants examined; Supplementary Table 2) and eukaryotic COS-7 cells (16 mutants examined; Supplementary Table 2). Though these mutations could not be secreted by COS-7 cells, they were still able to be efficiently secreted by E. coli cells. Each data point represents the relative secretion of a different mutant ($n = 3$ independent experiments). **e** Most part of CBS is buried within the native structure of human CRP. The crystal structure of CRP subunit (1B09)[56] is shown in both Ribbon (left) and Surface modes (right). CBS is colored in purple with the side chain of Cys36 also indicated. **f** The sequence logo of CBS generated with WebLogo 3[66]. Sequences of mouse, rat, and human CBS are also shown. The three pattern-decisive residues are highly conserved, whereas the rest are rather variable allowing fine adjustment of CBS activities. **g** IDPs (disorder contents of soluble protein or that of the extracellular region of membrane proteins >50%) in human (98), rodents (mouse and rat; 33), other multicellular organisms (drosophila, nematode, and arabidopsis; 22), and yeast (34) were retrieved from DisProt database[47] (Supplementary Table 3). Ratios of secretory IDPs with a signal peptide are much lower even when corrected for the ratio of secretory proteins versus the entire proteome (36%). **h** Ratios of disordered sequences in extracellular/luminal (non-cytoplasmic; median ratio = 0) versus cytoplasmic portions of all membrane IDPs (median ratio = 1) from all species regardless of their overall disordered contents or cellular localization (Supplementary Table 4; $n = 130$; left). The sequence lengths of the extracellular/luminal and cytoplasmic portions of these membrane IDPs are also shown (right). Disordered sequences are significantly enriched in cytoplasmic over extracellular/luminal portions of membrane proteins, whereas the overall lengths of the two portions are comparable. In box plots, centerline represents the median; box limits represent upper and lower quartiles; whiskers represent 1.5× interquartile range; points represent outliers. *$p < 0.05$; ***$p < 0.001$, Kolmogorov–Smironv tests, two-sided.

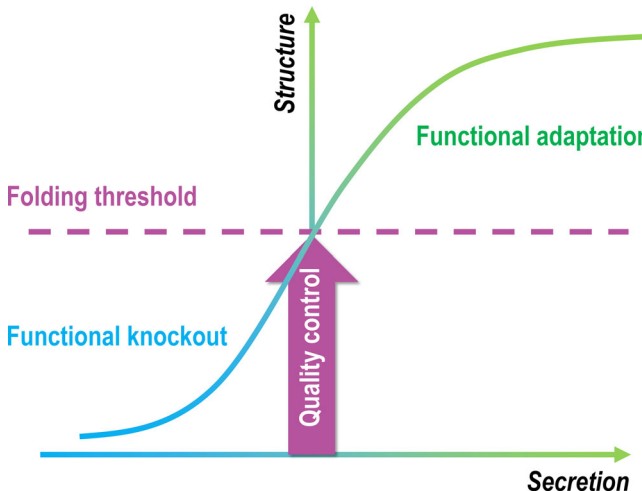

**Fig. 6 The proposed folding threshold model of protein evolution.** Functional constraints play a major role in shaping protein evolution, yet a folding threshold imposed by the eukaryotic quality control must be satisfied first before functional selection can be effective for a large portion of secretory and luminal proteins. Therefore, mutations that disrupt the original fold without giving rise to an alternative stable fold are equivalent to functional knockout. Only those that meet the threshold will be selected further leading to functional improvement and/or adaptation.

Regardless of whether levels or hidden activities change first, the conserved in vivo functions of CRP nevertheless argue that functional constraints are dominant over structural constraints in shaping CRP evolution. This evolutionary mode might also apply to a large portion of secretory and luminal proteins that are held in check by stringent eukaryotic quality control. By contrast, the contributions of structural constraints to protein evolution might be overestimated due to difficulties in identifying hidden activities and in relating levels and in vivo functions of orthologs individually. Therefore, selection on folding determinants is to pass the quality control, the success of which is the prerequisite for protein function. This per se constitutes a major functional constraint

and dictates the structural context for additional selection that fine-tunes protein activities. As such, it may be the maturation of quality control that limits structural innovation of proteins, thereby partly accounting for the vast mismatch between their sequences and structures[57–59].

## Methods

**Preparation of CRP.** CRP purified from human ascites (purity >97%) were purchased from the BindingSite (Birmingham, UK; catalog number: BP300.X). Coding sequences of mouse and rat CRP were cloned into pPic9k vectors with its N-terminus fused to a 6×His tag (molecular weight: ~0.93 kDa) and the signal peptide of α-factor. The vectors were transfected to P. pastoris strain GS115 cells for expression. CRP secreted into conditioned media were sequentially purified with HisTrap FF (BioCatal, Beijing, China; catalog number: 170901H5) and with p-Aminophenyl Phosphoryl Choline Gel (Thermo Fisher Scientific, Rockford, IL; catalog number: 20307; lot number: SF251770, TD268563).

**Characterization of CRP function in vivo.** CRP knockout (KO) mice of C57BL/6 background and CRP KO rats of SD background were generated by Shanghai Biomodel Organism Science & Technology Development (Shanghai, China) and Shanghai Model Organisms (Shanghai, China), respectively[30,60]. WT C57BL/6 mice and WT SD rats were obtained from the Animal Center of Xi'an Jiaotong University. Acute liver failure was induced in animals at age of 8~9 weeks by intraperitoneally (i.p.) injection of 300 (for mice) or 750 mg/kg (for rats) acet-aminophen (APAP) (Macklin, Shanghai, China; catalog number: A800441) after 12 h fasting. In rescue experiments, human CRP at the indicated dosage was i.p. administrated into WT mice at 2 h post APAP injection. Sera were sampled at 24 h post APAP injection, and circulating levels of alanine transaminase (ALT) were determined with kits (Nanjing Jiancheng Bioengineering Institute, Nanjing, China; catalog number: C009-2-1; lot number: 20190530). Sepsis was induced in animals at age of 8~9 weeks age by i.p. injection of 20 (for mice) or 8 mg/kg (for rats) LPS (Sigma-Aldrich, St. Louis, MO; catalog number: L4130; lot number: 099M4029). Survival was monitored thereafter. When indicated, 2.5 mg/kg human CRP was intravenously administrated into WT mice at the same time of LPS injection. All animal experiments complied with the Guide for the Care and Use of Laboratory Animals published by NIH, and were conducted according to the protocols approved by the Ethics Committee of Animal Experiments of Xi'an Jiaotong University (2020-423).

**Characterization of CRP structure and activities in vitro.** The structure of human, mouse, and rat CRP were examined with electron microscopy[32,39,61,62]. Briefly, samples were negatively stained with 1% uranyl acetate for 40 s and imaged with a FEI Talos F200C electron microscope at a magnification of ×120,000. Single-particle analysis was conducted using CryoSPARC[63] and RELION 2.0[64]. After CTF correction, a total of 5929 mouse CRP, 2873 rat CRP, and 6561 human CRP

particles were picked from 20, 11, and 12 high-quality micrographs, respectively. Unfolding of CRP (110 μg/ml) was induced by heating with a CFX96 Real-Time PCR Detection System (Bio-Rad, Hercules, CA) in the presence of SYPRO Orange (1:500; Sigma-Aldrich, St. Louis, MO; catalog number: S5692; lot number: MKCH9337). Fluorescence intensity was monitored to assess structural changes of CRP. Unfolding of CRP (100 μg/ml) was also induced by incubation with 8 M urea. Refolding of urea-unfolded CRP was induced by 100-fold dilution into TBS (20 mM Tris, 150 mM NaCl, pH 7.4) containing 2 mM CaCl$_2$ overnight. Calcium-dependent PC-binding activities were determined to assess the regeneration of native conformation.

For determining PC-binding activities of native CRP, 10 μg/ml PC-KLH (Santa Cruz Biotechnology, Dallas, TX; catalog number: sc-396490; lot number: J0915; Biosearch Technologies, Teddington; catalog number: PC-1013-5; lot number: 105269-03) was immobilized onto microtiter wells overnight. After blocking with TBS containing 1% BSA, CRP was added for 1 h in TBS containing 2 mM CaCl$_2$ or 5 mM EDTA, and binding was probed by anti-human CRP mAb 1D6 (1:200) and HRP-labeled goat-anti-mouse IgG (1:4000; Jackson ImmunoResearch, West Grove, PA; catalog number: 115-035-003; lot number: 125229) or by HRP-labeled anti-His antibody (1:10,000; Proteintech, Rosemont, IL; catalog number: 66005-I-Ig; lot number: 10004365). For determining activities of CBS, competitive ELISA against the binding of human CRP to 6 ligands (cholesterol, C1q, collagen, fibronectin, fibrinogen, ApoB) was performed[36,37]. Briefly, microtiter wells were coated with indicated ligands overnight at 4 °C and blocked with TBS containing 1% BSA. mCRP with or without the indicated CBS peptide at 80 μg/ml was then added for 1 h and detected with appropriate mAbs.

**Characterization of structural constraints of CRP**. WT CRP of distinct species was expressed in *E. coli* (BL21) cells[39]. Their coding sequences were fused to a strep tag and the signal peptide of alkaline phosphatase (ALP). After expression was induced for 24 h, cells and culture media were collected. Cells were lysed in buffers containing 1% Triton X-100, 20 mM Tris, 150 mM NaCl, 1 mM PMSF, pH 7.4 by pulse sonication on ice, and centrifugated at $5000 \times g$. The supernatants were subjected to immunoblotting to assess levels of soluble expression of CRP using an anti-strep mAb (1:30,000; Sangon Biotech, Shanghai, China; catalog number: D191106) and HRP-labeled goat-anti-mouse IgG (1:40,000). PC-binding activities in the supernatants and culture media were determined to assess levels of near folded and fully folded CRP. The supernatants and culture media were also heated at 70 °C for 10 min, and residue PC-binding activities were determined with ELISA to assess the stability of CRP using an anti-strep mAb (1:3000) and HRP-labeled goat-anti-mouse IgG (1:4000). The supernatants were treated with TBS containing 8 M Urea, 5 mM EDTA, 20 mM DTT followed by refolding with serial dialysis into regeneration buffers (10 mM Tris, 15 mM NaCl, 2 mM CaCl$_2$, 5% glycerol, pH 7.4) at decreasing concentrations of urea (6, 4, 2, and 0 M). PC-binding activities were then determined with ELISA to assess the foldability of CRP using an anti-strep mAb (1:3000) and HRP-labeled goat-anti-mouse IgG (1:4000).

**Characterization of folding determinants of CRP**. WT or mutant human CRP were expressed in *E. coli* (BL21) or COS-7 cells using pET42c or pcDNA3.1 vector as described in our previous work[39]. To enable secretion of CRP, a signal peptide of ALP or its own was fused to the coding sequence. The spontaneous core formation of CRP was examined with immunoblotting using 3H12 mAb (1:1000) and HRP-labeled goat-anti-mouse IgG (1:40,000) to assess the correct formation of the introduced inter-strand disulfide bond according to its migration pattern[39]. The subsequent nonspontaneous folding of CRP was examined with conformation-specific, sandwich ELISA to determine levels of secreted native CRP in culture media[32,39]. Briefly, microtiter wells were coated with a sheep-anti-human CRP polyclonal antibody (5 μg/ml; BindingSite; catalog number: DX044; lot number: 445029-1) to capture CRP in culture media. The captured native and non-native CRP was detected with 1D6 (1:200) and 3H12 mAbs (1:200), respectively. Total secretion of native and non-native CRP was examined with immunoblotting of culture media using 3H12 mAb (1:1000) and HRP-labeled goat-anti-mouse IgG (1:40,000).

**Statistics and reproducibility**. Data were presented as mean ± SEM. Data of in vivo experiments were obtained from 5 to 12 animals. Data of in vitro experiments were obtained from at least three independent replicates. The exact number of independent replicates is indicated in the figure legends. Statistical analysis was performed by two-tailed Student's *t* test, one-way analysis of variance with Tukey post hoc or Kolmogorov–Smirnov tests as appropriate. Values of $p < 0.05$ were considered significant.

**Reporting summary**. Further information on research design is available in the Nature Research Reporting Summary linked to this article.

## Data availability
The crystal structure of human CRP analyzed during the current study is available in the RCSB PDB database (1B09; www.rcsb.org). The sequences of CRP analyzed during the current study were obtained from the UniProt database (www.uniprot.org). The information of intrinsically disordered proteins was retrieved from the DisProt database (www.disprot.org). Single-particle datasets that support the findings of this study have been deposited in EMPIAR with accession codes EMPIAR-10960 (mouse CRP), EMPIAR-10959 (rat CRP), and EMPIAR-10958 (human CRP). The authors declare that all other data supporting the findings of this study are available within the article and its Supplementary Information files.

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

## Acknowledgements
This work was supported by grants from the National Natural Science Foundation of China [grant numbers 32071260, 31971186, and 31870767] and a grant from the Natural Science Foundation of Shaanxi Province [grant number 2020JQ-025].

## Author contributions
Y.W. and S.-R.J. designed the research. B.C., J.-M.L., Y.-L.L., L.Z., X.-P.H., and H.-Y.L. performed the research. Y.W., S.-R.J., L.A.P., B.C., and J.-M.L. analyzed the data and wrote the paper. All authors reviewed the results and approved the final version of the manuscript.

## Competing interests
The authors declare no competing interests.
