## [Peer Review File · Communications Biology]

Reviewers' comments:

Reviewer #1 (Remarks to the Author):

In this manuscript, the authors have investigated the evolutionary relationships among gene expression, structure, and functions of C-reactive protein (CRP) employing murine CRP, rat CRP and human CRP. The work is significant and the data are conclusive; however, the following points should be addressed:

1. The authors are directed to references 35-36 for the methods used to evaluate the binding of CRP to six ligands (other than PCh). Since two papers are referenced, it would be useful to include the methods, in brief, in this paper too.
2. Fig. 1C: Is there any specific reason why KO mice were not used for injection of human CRP?
3. Please provide the methods used for determining the KD values for the binding of CRP to PCh. Does the reported KD for human CRP-PCh match with the previously published KD for human CRP?
4. Fig. 3: The ligand-binding data for the six ligands are not shown; only the relative potency of various CRP species is shown. Please justify.
5. There are several typographical/grammatical mistakes in the manuscript.

Reviewer #3 (Remarks to the Author):

In the paper "Secretory quality control constrains functional selection-associated protein structure innovation" Cheng et al. take into consideration the role of functional constraints in the evolution of the CRP protein. The evolutionary proposed model can be divided into two steps, the first in which dominates the structural constraints, the second one in which, once satisfied with the quality control of secretion, can be selected mutations that improve the function or adaptation. I thought this study was potentially interesting and the authors made a good attempt at characterizing the constraints that dominate protein evolution.

Major comment:

-It's known that CRP can be found in two forms pentameric and monomeric with different roles in the inflammatory process, could be the difference of expression of pentameric CRP the result of differential efficiency in the pathway of proinflammatory cascade?

-Although the used statistical methods are declared, I could not find the number of replicate for each experiment both in vivo and in vitro, please add it in materials and method and the relative figure captions.

Minor comments

In Abstract

Line 4 "Selection on structural constraints is linked closely to levels of proteins", do the authors mean level of protein expression? Please specify

In Results

Line 17 "Given their drastic differences in levels" again do the authors mean level of protein expression? Please specify

Reviewer #4 (Remarks to the Author):

All page and line numbers listed in my comments are referring to the manuscript pdf file downloaded from the journal web site.

Secretory quality control constrains functional selection-associated protein structure innovation; Cheng et al. The authors have explored whether functional constraints are more important than structural constraints in shaping the evolution of a protein, and they have used C-reactive protein (CRP) as a test case. They have used in vivo animal models of acute liver failure and sepsis to demonstrate that mouse, rat and human CRP have a conserved function in protecting against tissue

injury arising from acute inflammation, despite quite different baseline circulating protein levels observed for the three species. The authors have then attempted to explain the circulating protein level-protein function mismatch observed across mouse, rat and human CRP by examining the 3-dimensional structural state of CRP and affinity for the native ligand phosphocholine (PC), coming to the conclusion that protein folding determinants play a major role in dictating the functional constraints of CRP. This is an interesting idea and worthy of publication, however there are several issues that the authors need to address. It is generally accepted that CRP exists in at least two conformational states, the circulating fully folded pentameric form (pCRP) and the largely unfolded monomeric form (mCRP). Circulating pCRP has been reported to be non-inflammatory and relatively inert, whereas mCRP is an activated pro-inflammatory conformational state of CRP (eg. Braig et al 2017 Nat Comms; Rajab et al 2020 Front Immunol; McFadyen et al 2018 Front Immunol; McFadyen et al 2020 Subcell Biochem; Potempa et al 2021 Front Immunol). It is widely thought that pCRP dissociates at the site of tissue injury or acute inflammation (eg. damaged cell membranes) into an activated pentameric state (usually denoted as pCRP* or mCRPm in the literature), exposing inflammatory neo-epitopes (eg. residues 199-206). The partially dissociated pentameric form of CRP (pCRP*) can then continue along the dissociation pathway to produce mCRP; both pCRP* and mCRP are pro-inflammatory. Detection of the different conformational states of CRP in in vitro and in vivo settings is usually carried out using conformation-specific anti-CRP monoclonal antibodies (mAbs); pCRP specific mAbs include 1D6, 8D8 and HD2.4, while 3H12 and 9C9 detect the neo-epitopes exposed in pCRP* and mCRP. The mAb CRP-8, widely used to detect CRP, has been reported to be specific for pCRP* and mCRP. Points to be addressed by the authors: 1. In the introduction, the authors should clarify that they are using “CRP” to define the circulating, non-inflammatory pentameric conformation of CRP and “unfolded CRP” to describe both pCRP* and mCRP. 2. The conformation-specific mAbs used in the study should be identified in the results and materials and methods sections – so that it is clear to the reader which conformation of CRP is being detected. It is not sufficient for the authors to merely cite their previous work, as they have done in the materials and methods section. The reader needs to understand which mAb detects which CRP conformation, without having to resort to previous published works. 3. The authors describe the “foldability” and “folding determinants” of CRP; these properties or characteristics are critical to their hypothesis, yet there is little detail in the main text as to what is actually meant by these terms and how they are calculated. To get an informed view, the reader has to turn to the authors previous publication in Scientific Reports (2018, Lv et al). Sufficient detail of these terms should be included in the results section of the manuscript. 4. The authors have used electron microscopy visualization, single particle analysis and homology modelling to show that mouse, rat and human pCRP adopt very similar 3-dimensional pentameric structures. They also demonstrate that the binding affinity (KD) of pCRP for PC-KLH is similar for all three species. Based on this, the authors conclude that there is little structural difference between pCRP for the three species and that structural differences cannot account for the observed differences in baseline circulating protein levels. Using differences in melting temperature (Tm), as well as the effects of 8 M urea, calcium and EDTA upon pCRP for the three species, the authors conclude that it is the ease of mCRP production that accounts for the variation in circulating pCRP levels across species. It is widely accepted by the authors and others (eg. Lv et al 2018 Sci Repts; Braig et al 2017 Nat Comms) that the first step in generating mCRP is the dissociation of the stable pentamer (ie. pCRP). The individual monomeric subunits within the stable human CRP pentamer are held together non-covalently by several critical salt bridges, as well as numerous hydrogen bonds and VDW interactions. It would therefore seem likely that the overall strength, and nature (ie. number of salt bridges, hydrogen bonds, VDW interactions), of the interactions between the individual subunits in pCRP will govern how readily the pentamer will dissociate for different species. However, the authors have not examined the inter-subunit

interactions in mouse, rat and human pCRP, either in silico or physically. At the very least, an in silico analysis of the inter-subunit interactions should be carried out to determine whether they are similar for the three species or if there are significant differences. This analysis would also inform the results presented in “Structural constraints play a minor role in shaping CRP evolution” and “Folding determinants dictate the native structure of CRP” sections.

5. In the results and discussion sections, the authors refer to the “drastic differences in levels” this should be more clearly described. One assumes that they are referring to the baseline circulating pCRP levels?

6. The regeneration of the native pCRP conformation was assessed using a binding assay to determine the “activity” of pCRP for immobilized PC-KLH. The authors also use a competitive ELISA assay to determine the “activity” of the mCRP CBS motif for six different ligands (cholesterol, C1q, collagen, fibronectin, fibrinogen, ApoB). Both of these are binding assays (ie. measuring the affinity of the ligand to the protein) not functional assays – to describe the affinity as activity or potency is incorrect and the authors should amend their description in the text and figures. Also, it would be beneficial for a little more detail to be included in the method section for the competitive ELISA assay rather than simply referring the reader to their previous work.

7. In Figure 3E, the dose of 5 mg/kg of huCRP is no longer protective in the mouse liver failure model and a reason for this is proposed on page 6, lines 6-9. Why didn’t the authors use conformation specific mAbs to determine whether pCRP, pCRP* and/or mCRP were present in the mouse serum for the 3 doses of huCRP? One would expect that levels of pCRP* and/or mCRP would be higher than that of pCRP for the 5 mg/kg dose compared to the 1.25 and 2.5 mg/kg doses.

8. On page 6, line 36: the authors state that the PC-binding site on the recognition face of pCRP is significantly more variable across species (Figure 5A). Yet, in Figure 2C the affinities for PC-KLH are very similar for mouse, rat and human pCRP and the authors acknowledge this on page 5, lines 23-24. The panel in Figure 5A shows that the PC-binding site is well conserved (ie. mostly colored white and shades of magenta) – the authors should amend their text on page 6 accordingly.

9. On page 7, first paragraph: the authors refer to β strands A to M. Yet only β strands C to H are labeled in Figure 5B, it would be helpful for all of the β strands of a pCRP subunit to be labeled in a supplementary figure. In addition, this paragraph describes a model for the pCRP folding process published by the authors previously (Lv et al 2018 Sci Reps), however it is impossible to understand the data being presented in the paragraph without prior reading of the Scientific Reports article. More detail should be added to the start of the paragraph about the pCRP folding model, so that the reader has a basic understanding of the model without the need to go to the Scientific Reports article. Alternatively, the authors could describe their existing CRP folding model in the introduction section of the manuscript.

10. The authors should make it clear that data for some of the mutants in Tables S1 and S2 has been previously reported in the Scientific Reports article (Lv et al 2018), this could be done (for example) by adding a superscript to these mutants in the supplementary tables.

11. In Table S4, the authors have included proteins with either no non-cytoplasmic region or an extremely short non-cytoplasmic region (ie. < 10 residues in length). It is unlikely that extremely short regions will have an ordered secondary structure. Including proteins without a non-cytoplasmic region or with an extremely short noncytoplasmic region will surely skew the analysis of the disordered ratio (Figure 5H). Can the authors comment on why these proteins were included in their analysis? What happens if these proteins are removed from their analysis?

12. First paragraph of the discussion, page 8, lines 11-14: As stated above, it is generally accepted that pCRP is non-inflammatory and relatively inert. The pro-inflammatory conformations of CRP are pCRP* and mCRP and their role/presence/importance in human disease and injury (eg. burns) has been previously reported (refer to references cited above). The final sentence of the first paragraph should be amended accordingly to reflect this.

13. Second paragraph of the discussion, page 8. The question of why CRP has kept the pentameric conformation during evolution has been raised previously in several publications by L. A Potempa. He proposes that the stable non-inflammatory form of pCRP

provides a delivery mechanism to get mCRP to the site of injury or inflammation. Potempa's hypothesis should be included in the discussion here. Minor points to be addressed: 1. In Figure 5 and in the main text: cytoplasmic often has a typographical error and is written as "cytoplamsic". ELISA has also been mistyped on page 11 in the materials and methods section. These errors should be corrected. 2. Page 10, line 5: the size of the His tag should be included. 3. Page 10, line 19: the time unit is missing "WT mice at 2 post APAP...". I assume that it is supposed to be 2 h? 4. Page 10, line 36: it is not sufficient to describe a buffer as "urea-free", the buffer composition and pH should be included in brackets, so that it reads "urea-free buffer (composition, pH) overnight." Buffer composition and pH should also be added to page 11, line 18. 5. Page 11, line 5: typographical error "precious work" should read as "previous work". 6. Page 11, lines 12, 24 and 28: the conformation specific CRP antibodies used for the immunoblotting should be listed. 7. In the legends for Figures 1 - 5, the number of independent experiments and replicates should be given (where practicable), along with what is being presented eg. data is expressed as the mean + SEM of n = 4 independent experiments carried out in duplicate. 8. Figure 2B: the number of degrees of rotation (ie. 180o) should be inserted next to (or below) the arrow. 9. Where PDB codes are given for a protein structure eg. (PDB 1B09), the original reference for the structure should also be given and included in the reference list. 10. Figure 4: what are the units for the Y-axes in panels B-G? 11. Figure 5B legend: the authors should include a citation for reference 39 (Lv et al 2018 Sci Reps). 12. Figure 5E: it is very hard to see Cys36 in the structure, to make the location of the residue clear it should be indicated by an arrow and labeled. 13. Authors should consider rewording the title – the current title is hard to understand and fails to mention CRP at all. The running title is more to the point, perhaps incorporating CRP into the running title would be a better option?

Response to Reviewer 1:

We greatly appreciate the helpful and constructive comments. Below we reiterate your concerns and provide our responses.

- 1. The authors are directed to references 35-36 for the methods used to evaluate the binding of CRP to six ligands (other than PCh). Since two papers are referenced, it would be useful to include the methods, in brief, in this paper too.**

We apologize for the oversight. The methods used to evaluate the binding of unfolded CRP, *i.e.* mCRP, to six ligands have been included in the revised manuscript (page 11, line 9 - 11).

- 2. Fig. 1C: Is there any specific reason why KO mice were not used for injection of human CRP?**

Injection of human CRP into CRP KO mice will restore the CRP-normal situation, which is certainly helpful to clarify whether mouse and human CRP are functionally equivalent. In this manuscript, mice of CRP KO, wildtype, and wildtype with human CRP injection were used to model CRP-deficient, CRP-normal, and CRP-overexpressed situations, respectively. The phenotypes of CRP in acute inflammation were found to be consistent across the three situations. Therefore, these results also demonstrate a conserved *in vivo* function of CRP. Moreover, an additional advantage of injection of huCRP into wildtype mice is that it could better reflect the therapeutic potential of CRP.

- 3. Please provide the methods used for determining the KD values for the binding of CRP to PCh. Does the reported KD for human CRP-PCh match with the previously published KD for human CRP?**

We apologize for the oversight. The KD values were determined by ELISA wherein dose-dependent binding of CRP to immobilized PC-KLH were measured. This information has been indicated in the revised manuscript (page 10, line 41 - page 11, line 6). In a previous study using a similar approach, the half-maximal binding of human CRP to PC-BSA and PnC occurred at concentrations of around 300 and 50 ng/ml¹, respectively, the latter of which is comparable to KD values determined herein (0.43 nM or 49.4 ng/ml). The differences in affinity might be partly ascribed to variations

in PC densities present by the ligands.

4. Fig. 3: The ligand-binding data for the six ligands are not shown; only the relative potency of various CRP species is shown. Please justify.

We apologize for lacking detailed descriptions in the first version of our manuscript. The hidden activities were evaluated by competitive ELISA using CBS peptide at a concentration of 80 $\mu\text{g/ml}$ to compete mCRP binding to ligands. Six ligands in total were examined. Relative potency for each ligand was defined as: binding of mCRP in the absence of CBS divided by that in the presence of CBS. Therefore, in Figure 3d, each bar is calculated using 6 data points, and each data point represents a potency index for a different ligand calculated based on three independent replicates. This information has been indicated in the revised manuscript (page 19, line 10 - 12).

5. There are several typographical/grammatical mistakes in the manuscript.

We apologize for these mistakes and have carefully revised the manuscript.

Response to Reviewer 3:

We greatly appreciate the helpful and constructive comments. Below we reiterate your concerns and provide our responses.

- 1. It's known that CRP can be found in two forms pentameric and monomeric with different roles in the inflammatory process, could be the difference of expression of pentameric CRP the result of differential efficiency in the pathway of proinflammatory cascade?**

As suggested by the reviewer, it is indeed possible that the difference of expression of pentameric CRP is the result of differential efficiency in the pathway of proinflammatory cascade. Nevertheless, it should be noted that the acute phase expression of human CRP can be recapitulated in transgenic mice harboring both coding and noncoding regulatory sequences of human CRP ^{2, 3}. We thus favor the explanation that variations in expression pattern of CRP across species are caused by mutations in noncoding regulatory sequences, following which mutations to coding sequences are selected to maintain a conserved functional phenotype by adjusting the hidden activities.

- 2. Although the used statistical methods are declared, I could not find the number of replicate for each experiment both in vivo and in vitro, please add it in materials and method and the relative figure captions.**

We apologize for lacking detailed descriptions in the first version of our manuscript. We have carefully revised the manuscript to include the required information (page 11, line 42 - page 12, line 1).

- 3. Line 4 "Selection on structural constraints is linked closely to levels of proteins", do the authors mean level of protein expression? Please specify**

The term "mRNA level" is used in the original work ⁴, while the term "protein abundance" is more frequently used thereafter particularly when discussing structural constraints ^{5, 6}. Nevertheless, mRNA levels are positively correlated with protein abundance in most cases. Therefore, the two terms can be used here interchangeably. We have specified this as expression levels in the revised manuscript.

4. In Result Line 17 “Given their drastic differences in levels” again do the authors mean level of protein expression? Please specify.

In the case of CRP, the protein abundance (circulating levels) is primarily determined by its mRNA levels ⁷. Therefore, we have specified this as circulating levels in the revised manuscript.

Response to Reviewer 4:

We greatly appreciate the helpful and constructive comments. Below we reiterate your concerns and provide our responses.

- 1. In the introduction, the authors should clarify that they are using “CRP” to define the circulating, non-inflammatory pentameric conformation of CRP and “unfolded CRP” to describe both pCRP* and mCRP.**

We thank for the suggestion of the reviewer, and have clearly indicated in the Introduction that “unfolded CRP encompasses conformational states denoted as mCRP, mCRPm, and pCRP* in literatures” (page 4, line 4-5). In some context CRP refers to isoform with native structure, while in other context it describes all isoforms such as in mentioning biological functions or evolution. We therefore carefully revised the manuscript to avoid confusion.

- 2. The conformation-specific mAbs used in the study should be identified in the results and materials and methods sections – so that it is clear to the reader which conformation of CRP is being detected. It is not sufficient for the authors to merely cite their previous work, as they have done in the materials and methods section. The reader needs to understand which mAb detects which CRP conformation, without having to resort to previous published works.**

We apologize for lacking detailed descriptions in the first version of our manuscript. In our experiments, 1D6 and 8D8 mAbs were used to detect the native conformation of CRP, whereas 3H12 mAb was used to detect the unfolded conformation⁸. We have carefully revised the manuscript to include the required information.

- 3. The authors describe the “foldability” and “folding determinants” of CRP; these properties or characteristics are critical to their hypothesis, yet there is little detail in the main text as to what is actually meant by these terms and how they are calculated. To get an informed view, the reader has to turn to the authors previous publication in Scientific Reports (2018, Lv et al). Sufficient detail of these terms should be included in the results section of the manuscript.**

We apologize for lacking detailed descriptions in the first version of our manuscript. Foldability refers to the efficiency of a protein to fold to its native conformation⁹,

whereas folding determinants refer to residues with significant impact on the foldability of a protein¹⁰. Foldability of a protein can be assessed in cells by expressing it to determine levels of soluble species in cell lysates with immunoblotting, levels of native conformation in cell lysates with ELISA, and levels of native conformation secreted into cell culture media with ELISA^{9, 11}. Foldability of a protein can also be assessed by examining the regeneration of native conformation after denaturation with ELISA. We have carefully revised the manuscript to include the required information (page 6, line 17; page 7, line 11).

4. The authors have used electron microscopy visualization, single particle analysis and homology modelling to show that mouse, rat and human pCRP adopt very similar 3-dimensional pentameric structures. They also demonstrate that the binding affinity (KD) of pCRP for PC-KLH is similar for all three species. Based on this, the authors conclude that there is little structural difference between pCRP for the three species and that structural differences cannot account for the observed differences in baseline circulating protein levels. Using differences in melting temperature (Tm), as well as the effects of 8 M urea, calcium and EDTA upon pCRP for the three species, the authors conclude that it is the ease of mCRP production that accounts for the variation in circulating pCRP levels across species. It is widely accepted by the authors and others (eg. Lv et al 2018 Sci Reps; Braig et al 2017 Nat Comms) that the first step in generating mCRP is the dissociation of the stable pentamer (ie. pCRP). The individual monomeric subunits within the stable human CRP pentamer are held together non-covalently by several critical salt bridges, as well as numerous hydrogen bonds and VDW interactions. It would therefore seem likely that the overall strength, and nature (ie. number of salt bridges, hydrogen bonds, VDW interactions), of the interactions between the individual subunits in pCRP will govern how readily the pentamer will dissociate for different species. However, the authors have not examined the inter-subunit interactions in mouse, rat and human pCRP, either in silico or physically. At the very least, an in silico analysis of the inter-subunit interactions should be carried out to determine whether they are similar for the three species or if there are significant differences. This analysis would also inform the results presented in “Structural constraints play a minor role in shaping CRP evolution” and “Folding determinants dictate the native structure of CRP” sections.

We have performed *in silico* analysis as suggested by the reviewer. According to the crystal structure of human CRP and modeled structures of mouse and rat CRP, the three inter-subunit salt bridges supposed to mediate the assembly of human CRP (D155-R118, E197-K123, K201-E101)¹² are present exactly in mouse (D155-R118, D197-K123, K201-E101) and rat CRP (D153-R116, D195-K121, K199-E99). Further

analysis was performed to detect possible inter-subunit ionic and hydrophobic interactions with Protein Interactions Calculator ¹³, and to detect possible inter-subunit hydrogen bonds with UCSF Chimera. The results are shown in Table I below. Mouse, rat, and human CRP harbor 25, 21, and 23 inter-subunit interactions, respectively, and no major difference could be noted among these inter-subunit interactions. Therefore, these inter-subunit interactions cannot explain the observed differences in the ease of CRP dissociation.

We have recently experimentally determined the structure of mouse CRP at atomic resolution. The structural data of rat CRP has also been collected and is being actively resolved. With these two structures (which are more accurate than the modeled structures), the concerns of the reviewer might be addressed satisfactorily in our future manuscript.

Table I. Interactions between two adjacent subunits in pentameric CRP

	Mouse CRP	Rat CRP	Human CRP
Ionic	5	7	6
Hydrophobic	10	7	8
Hydrogen bond	10	7	9

5. In the results and discussion sections, the authors refer to the “drastic differences in levels” this should be more clearly described. One assumes that they are referring to the baseline circulating pCRP levels?

We apologize for the ambiguous descriptions. Under the baseline condition, the circulating levels of CRP in three species differ most. Under inflammatory conditions, the circulating levels of CRP in three species also show large differences. The circulating levels of human CRP may approach that of rat CRP (300 - 500 µg/ml) under somewhat extreme situations. We have carefully revised the manuscript to clearly indicate that the differences are referring to the circulating levels.

6. The regeneration of the native pCRP conformation was assessed using a binding assay to determine the “activity” of pCRP for immobilized PC-KLH. The authors also use a competitive ELISA assay to determine the “activity” of the mCRP CBS motif for six different ligands (cholesterol, C1q, collagen, fibronectin, fibrinogen, ApoB). Both of these are binding assays (ie. measuring the affinity of the ligand to the protein) not functional assays – to describe the affinity as activity or potency is incorrect and the authors

should amend their description in the text and figures. Also, it would be beneficial for a little more detail to be included in the method section for the competitive ELISA assay rather than simply referring the reader to their previous work.

We apologize for the ambiguous descriptions and the lack of details. We fully agree the reviewer that binding is not explicitly equal to function, though these two are closely associated. In this manuscript, we have dissociated the terms of activity and function. Activity measures how active a protein manifests *in vitro* by using binding assays, while function measures how important a protein is *in vivo* by using phenotypic assays. Activity is the basis of function, yet expression level constitutes another key determinant of function. In evolutionary studies, catalytic activities or binding affinities to ligands or receptors measured *in vitro* are frequently used to infer the functions of proteins *in vivo*⁹. However, expression levels are usually ignored in previous studies.

For CRP, calcium-dependent PC binding is the most established activity, and is considered as the basis underlying many of its putative functions including pattern recognition and complement regulation. For unfolded CRP, the markedly enhanced ligand binding activities appear also to be closely associated with its putative functions. We thus used binding assays to measure its *in vitro* activities. As pointed by the reviewer, binding activities and function of a protein are not equal. We fully agree with the reviewer, but also acknowledge that when ignoring expression levels, binding activities can, to certain extent, be used as a functional index.

The manuscript has been carefully revised according to the comments of the reviewer (page 5, line 23-24).

7. In Figure 3E, the dose of 5 mg/kg of huCRP is no longer protective in the mouse liver failure model and a reason for this is proposed on page 6, lines 6-9. Why didn't the authors use conformation specific mAbs to determine whether pCRP, pCRP* and/or mCRP were present in the mouse serum for the 3 doses of huCRP? One would expect that levels of pCRP* and/or mCRP would be higher than that of pCRP for the 5 mg/kg dose compared to the 1.25 and 2.5 mg/kg doses.

We thank for the suggestion of the reviewer. The determination of mCRP levels following injection of different doses of CRP into mice with ongoing liver injury should indeed be very informative and helpful to our proposal. However, there are several technical difficulties regarding mCRP detection in mice sera. In particular, the half-life of CRP is less than 4 h and the generated mCRP tends to be tissue-localized^{14, 15}. Therefore, without continuous supply of CRP, the levels of mCRP leaked into the circulation would be exceptionally low. Indeed, in a simplified model of mCRP

detection, 12 h following a single *i.p.* injection of CRP at 5 mg/kg into LPS-treated mice, levels of mCRP in peritoneal fluids were around 0.3 ng/ml, almost below the detection limit of our ELISA developed to detect mCRP in biological samples ¹⁶, and showed large variations (Figure I). Therefore, we were unable to reliably detect and compare mCRP levels in our models. Extensive optimization or new approach appears to be required for this objective.

Figure I. Levels of mCRP in peritoneal fluids of mice with sepsis. in Wildtype mice were treated with 10 mg/kg LPS and 5 mg/kg CRP intraperitoneally. Peritoneal fluids were collected at 12 h. Levels of CRP (left; capture Ab: BindingSite pAb, detection Ab: 1D6) and mCRP (right; capture Ab: Clone 8, detection Ab: 3H12) were detected by conformation-specific ELISA ¹⁶.

8. On page 6, line 36: the authors state that the PC-binding site on the recognition face of pCRP is significantly more variable across species (Figure 5A). Yet, in Figure 2C the affinities for PC-KLH are very similar for mouse, rat and human pCRP and the authors acknowledge this on page 5, lines 23-24. The panel in Figure 5A shows that the PCbinding site is well conserved (ie. mostly colored white and shades of magenta) – the authors should amend their text on page 6 accordingly.

The PC-binding site of human CRP comprises 4 residues, *i.e.* L64, F66, T76, and E81 ¹. The conservation scores of these residues are -0.743, -0.001, 0.589, and 0.392, respectively. They are significantly more variable than the 7 folding-decisive residues with a conservation score of -1.122 ± 0.106 (mean \pm SEM) ($p = 0.0013$, two-tailed Student's t-test). By contrast, they do not differ from the other 12 binding-decisive residues with a conservation score of 0.488 ± 0.266 (mean \pm SEM) ($p = 0.402$, two-tailed Student's t-test). Moreover, L64 is the most conserved one of the 4 PC-binding residues, yet its importance in PC binding has not been confirmed ¹. By contrast, the less conserved F66 have been demonstrated to be the major binding determinant ¹.

Though the PC-binding residues are less conserved, 3 of the 4 PC-binding residues (L64, F66, and E81) are identical in mouse, rat, and human CRP (Figure S2b), which can explain their comparable PC-binding affinities (Figure 4c). These together indicate that PC binding activities associated with the native structure of CRP cannot explain their level-function mismatch, and therefore do not constitute the functional constraints.

- 9. On page 7, first paragraph: the authors refer to β strands A to M. Yet only β strands C to H are labeled in Figure 5B, it would be helpful for all of the β strands of a pCRP subunit to be labeled in a supplementary figure. In addition, this paragraph describes a model for the pCRP folding process published by the authors previously (Lv et al 2018 Sci Reps), however it is impossible to understand the data being presented in the paragraph without prior reading of the Scientific Reports article. More detail should be added to the start of the paragraph about the pCRP folding model, so that the reader has a basic understanding of the model without the need to go to the Scientific Reports article. Alternatively, the authors could describe their existing CRP folding model in the introduction section of the manuscript.**

We thank for the suggestion of the reviewer, and have carefully revised the manuscript to include the necessary background information. In Figure II below and Figure S1 in the revised manuscript shows CRP subunit with all the β strands labeled, and the model for native structure formation of CRP.

Step 1: conformation folding-driven, spontaneous

Step 2: disulfide bonding-driven, non-spontaneous

Figure II. The model for native structure formation of CRP. Upper panel: subunit structure of human CRP (PDB 1B09)¹² with all β strands labeled. Lower panel: the folding and assembly of CRP subunits¹⁷. Step 1 of subunit folding is a spontaneous process: strands A & B folds first, leading to the formation of C-I core; independently folded 168-176 helix then anchors to C-I core covering Cys36 and Cys97, eventually resulting in their bonding. At this step, strands B & L are mislocated to the edge of C-I core. Step 2 of subunit folding is a non-spontaneous process requiring the assistance of cellular factors. In this process, disulfide bonding evokes

conformation changes of C-I core, leading to relocation of strands B & L for integration of strands J & K coordinated by calcium binding. Finally, folded subunits are assembled into the pentamer.

10. The authors should make it clear that data for some of the mutants in Tables S1 and S2 has been previously reported in the Scientific Reports article (Lv et al 2018), this could be done (for example) by adding a superscript to these mutants in the supplementary tables.

We thank for the reminder of the reviewer, and have carefully revised the manuscript as suggested.

11. In Table S4, the authors have included proteins with either no non-cytoplasmic region or an extremely short non-cytoplasmic region (ie. < 10 residues in length). It is unlikely that extremely short regions will have an ordered secondary structure. Including proteins without a non-cytoplasmic region or with an extremely short noncytoplasmic region will surely skew the analysis of the disordered ratio (Figure 5H). Can the authors comment on why these proteins were included in their analysis? What happens if these proteins are removed from their analysis?

We thank for the suggestion of the reviewer and have performed the suggested analysis. Of the 130 membrane IDPs analyzed, 22 harbor non-cytoplasmic regions less than 10 residues in length. Unexpectedly, 20 of the 22 proteins have their non-cytoplasmic regions completely ordered (Table II). This suggests an even more stringent quality control on shorter non-cytoplasmic sequences to keep their stability and function. After excluding these proteins, disordered sequences are still significantly enriched in cytoplasmic over extracellular/luminal portions of membrane IDPs, whereas the overall lengths of non-cytoplasmic sequences are a bit longer (Figure III).

Additional analysis revealed that membrane IDPs with shorter non-cytoplasmic regions tends to be less disordered, whereas membrane proteins with shorter cytoplasmic regions tends to be more disordered (Figure IV). This also suggest that the cytoplasmic and non-cytoplasmic regions undergo distinct quality control.

Table II. Intrinsically disordered membrane proteins with non-cytoplasmic regions less than 10 residues

Protein *	Disordere d content	Ratio of cytoplasmic	Ratio of non-	Length of cytoplamsi	Length of non-
-----------	------------------------	-------------------------	------------------	-------------------------	-------------------

		c disorder	cytoplamsi c disorder	c portion	cytoplasmic c portion
T-cell surface glycoprotein CD3 zeta chain	83.54%	100.00%	100.00%	113	9
Linker for activation of T- cells family member 1	78.63%	87.66%	0.00%	235	4
Vesicle- associated membrane protein- associated protein B/C	51.44%	56.11%	0.00%	221	0
Bcl-2-like protein 1	39.48%	44.02%	0.00%	209	7
Stannin	29.55%	45.61%	0.00%	57	10
High affinity immunoglobuli n epsilon receptor subunit gamma	24.42%	38.10%	0.00%	42	5
Vesicle- associated membrane protein 2	24.14%	30.11%	0.00%	93	2
Protein phosphatase 1 regulatory subunit 15A	17.66%	18.14%	0.00%	656	0
Mitochondrial fission 1 protein	16.45%	1.64%	22.22%	122	9
Apoptosis regulator Bcl-2	9.62%	10.90%	0.00%	211	6
Vesicle- associated membrane	50.35%	60.17%	0.00%	118	2

protein 4					
Vesicle-associated membrane protein 2	75.86%	93.55%	0.00%	93	2
Vesicle transport through interaction with t-SNAREs homolog 1A	34.82%	39.20%	0.00%	199	4
Syntaxin-12	25.18%	27.71%	0.00%	249	3
Bcl-2-like protein 1	21.46%	23.92%	0.00%	209	7
Syntaxin-1A	7.64%	8.30%	0.00%	265	0
Synaptobrevin homolog 1	79.49%	98.94%	0.00%	94	6
Synaptobrevin homolog 2	79.13%	97.85%	0.00%	93	3
Protein SSO1	25.52%	27.92%	0.00%	265	3
Protein SSO2	25.08%	27.51%	0.00%	269	4
Translocase of chloroplast 132, chloroplastic	37.73%	38.53%	0.00%	1181	7
Protein phosphatase 2C 70	2.58%	2.71%	0.00%	553	7

Figure III. Cytoplasmic portions of membrane IDPs are more disordered. Membrane IDPs with non-cytoplasmic regions longer than 10 residues were analyzed ($n = 108$). Left: ratios of disordered sequences in extracellular/luminal (non-cytoplasmic) versus cytoplasmic portions of. Right: sequence lengths of the extracellular/luminal and cytoplasmic portions. p values were determined by Kolmogorov-Smirnov tests.

Figure IV. Shorter non-cytoplasmic portions of membrane IDPs are less disordered. Membrane IDPs ($n = 130$) were divided into two halves according to lengths of their non-cytoplasmic (a) or cytoplasmic portions (b). Left: ratios of disordered sequences in extracellular/luminal (non-cytoplasmic) portions. Right: ratios of disordered sequences in cytoplasmic portions. p values were determined by

Kolmogorov-Smirnov tests.

12. First paragraph of the discussion, page 8, lines 11-14: As stated above, it is generally accepted that pCRP is non-inflammatory and relatively inert. The pro-inflammatory conformations of CRP are pCRP* and mCRP and their role/presence/importance in human disease and injury (eg. burns) has been previously reported (refer to references cited above). The final sentence of the first paragraph should be amended accordingly to reflect this.

We have carefully revised the manuscript to include the previous work as suggested by the reviewer (page 8, line 12-14).

13. Second paragraph of the discussion, page 8. The question of why CRP has kept the pentameric conformation during evolution has been raised previously in several publications by L. A Potempa. He proposes that the stable non-inflammatory form of pCRP provides a delivery mechanism to get mCRP to the site of injury or inflammation. Potempa's hypothesis should be included in the discussion here.

We thank for the reminder of the reviewer, and have carefully revised the manuscript to include the hypothesis of Dr. Potempa (page 8, line 17-19).

14. In Figure 5 and in the main text: cytoplasmic often has a typographical error and is written as "cytoplamsic". ELISA has also been mistyped on page 11 in the materials and methods section. These errors should be corrected.

We apologize for the oversight, and have carefully revised the manuscript to correct these errors.

15. Page 10, line 5: the size of the His tag should be included.

We apologize for the oversight. The size of His tag (6×His, ~0.93 kDa) has been included in the revised manuscript (page 10, line 5).

16. Page 10, line 19: the time unit is missing "WT mice at 2 post APAP...".

assume that it is supposed to be 2 h?

We apologize for the oversight. The missing time unit has been included in the revised manuscript (page 10, line 20).

17. Page 10, line 36: it is not sufficient to describe a buffer as “urea-free”, the buffer composition and pH should be included in brackets, so that it reads “urea-free buffer (composition, pH) overnight.” Buffer composition and pH should also be added to page 11, line 18.

We apologize for lacking detailed descriptions in the first version of our manuscript. We have carefully revised the manuscript to include the necessary information (page 10, line 37-38).

18. Page 11, line 5: typographical error “precious work” should read as “previous work”.

We apologize for the oversight, and have carefully revised the manuscript to correct these errors

19. Page 11, lines 12, 24 and 28: the conformation specific CRP antibodies used for the immunoblotting should be listed.

We apologize for lacking detailed descriptions in the first version of our manuscript. We have carefully revised the manuscript to include the necessary information (page 11, line 19-20, line 24, line 28).

20. In the legends for Figures 1 - 5, the number of independent experiments and replicates should be given (where practicable), along with what is being presented eg. data is expressed as the mean + SEM of n = 4 independent experiments carried out in duplicate.

We apologize for lacking detailed descriptions in the first version of our manuscript. We have carefully revised the manuscript to include the necessary information.

21. Figure 2B: the number of degrees of rotation (ie. 180o) should be inserted next to (or below) the arrow.

We thank for the suggestion of the reviewer, and have revised Figure 2b to include the degree of rotation (Figure V).

Figure V. Revised Figure 2b.

22. Where PDB codes are given for a protein structure eg. (PDB 1B09), the original reference for the structure should also be given and included in the reference list.

We thank for the reminder of the reviewer, and have included the references in the revised manuscript.

23. Figure 4: what are the units for the Y-axes in panels B-G?

The Y-axes in Figure 4b-g are in relative values normalized to that of human CRP. The legends of these figures have been revised to clearly indicate this (page 21, line 10-11).

24. Figure 5B legend: the authors should include a citation for reference 39 (Lv et al 2018 Sci Reps).

We thank for the reminder of the reviewer, and have included this references in the revised manuscript.

25. Figure 5E: it is very hard to see Cys36 in the structure, to make the location of the residue clear it should be indicated by an arrow and labeled.

We thank for the suggestion of the reviewer, and have revised Figure 5e as suggested (Figure VI).

Figure VI. Revised Figure 5e.

26. Authors should consider rewording the title – the current title is hard to understand and fails to mention CRP at all. The running title is more to the point, perhaps incorporating CRP into the running title would be a better option?

We thank for the suggestion of the reviewer, and the running title has been revised to include CRP, *i.e.* “Functional constraints dominate CRP evolution”.

References:

1. Agrawal A, Simpson MJ, Black S, Carey MP, Samols D. A c-reactive protein mutant that does not bind to phosphocholine and pneumococcal c-polysaccharide. *J Immunol.* 2002;169:3217-3222
2. Murphy C, Beckers J, Ruther U. Regulation of the human c-reactive protein gene in transgenic mice. *J Biol Chem.* 1995;270:704-708
3. Szalai AJ, van Ginkel FW, Dalrymple SA, Murray R, McGhee JR, Volanakis JE. Testosterone and il-6 requirements for human c-reactive protein gene expression in transgenic mice. *J Immunol.* 1998;160:5294-5299
4. Pal C, Papp B, Hurst LD. Highly expressed genes in yeast evolve slowly. *Genetics.* 2001;158:927-931
5. Zhang J, Yang JR. Determinants of the rate of protein sequence evolution. *Nat Rev Genet.* 2015;16:409-420

6. Echave J, Wilke CO. Biophysical models of protein evolution: Understanding the patterns of evolutionary sequence divergence. *Annu Rev Biophys.* 2017;46:85-103
7. Pepys MB, Hirschfield GM. C-reactive protein: A critical update. *J Clin Invest.* 2003;111:1805-1812
8. Ying SC, Gewurz H, Kinoshita CM, Potempa LA, Siegel JN. Identification and partial characterization of multiple native and neoantigenic epitopes of human c-reactive protein by using monoclonal antibodies. *J Immunol.* 1989;143:221-228
9. Smock RG, Yadid I, Dym O, Clarke J, Tawfik DS. De novo evolutionary emergence of a symmetrical protein is shaped by folding constraints. *Cell.* 2016;164:476-486
10. Lv JM, Chen JY, Liu ZP, Yao ZY, Wu YX, Tong CS, Cheng B. Cellular folding determinants and conformational plasticity of native c-reactive protein. *Front Immunol.* 2020;11:583
11. Zheng J, Guo N, Wagner A. Selection enhances protein evolvability by increasing mutational robustness and foldability. *Science.* 2020;370
12. Thompson D, Pepys MB, Wood SP. The physiological structure of human c-reactive protein and its complex with phosphocholine. *Structure.* 1999;7:169-177
13. Tina KG, Bhadra R, Srinivasan N. Pic: Protein interactions calculator. *Nucleic Acids Res.* 2007;35:W473-476
14. Motie M, Schaul KW, Potempa LA. Biodistribution and clearance of ¹²⁵I-labeled c-reactive protein and ¹²⁵I-labeled modified c-reactive protein in cd-1 mice. *Drug Metabolism & Disposition.* 1998;26:977-981
15. Li H, Wang J, Wu Y, Zhang L, Liu Z, Filep J, Potempa L, Wu Y, Ji S. Topological localization of monomeric c-reactive protein determines proinflammatory endothelial cell responses. *Journal of Biological Chemistry.* 2014;289:14283-14290
16. Zhang L, Li HY, Li W, Shen ZY, Wang YD, Ji SR, Wu Y. An elisa assay for quantifying monomeric c-reactive protein in plasma. *Front Immunol.* 2018;9:511
17. Lv JM, Lu SQ, Liu ZP, Zhang J, Gao BX, Yao ZY, Wu YX, Potempa LA, Ji SR, Long M, Wu Y. Conformational folding and disulfide bonding drive distinct stages of protein structure formation. *Sci Rep.* 2018;8:1494

REVIEWERS' COMMENTS:

Reviewer #1 (Remarks to the Author):

none

Reviewer #4 (Remarks to the Author):

In the revised manuscript and rebuttal document, the Authors have addressed all of the points raised by the three reviewers in great detail. Therefore, the manuscript is highly recommended for publication in Communications Biology.